# Metabolic Mechanisms Underlying Heat and Drought Tolerance in Lentil Accessions: Implications for Stress Tolerance Breeding

**DOI:** 10.3390/plants12233962

**Published:** 2023-11-24

**Authors:** Noureddine El Haddad, Youness En-nahli, Hasnae Choukri, Khawla Aloui, Rachid Mentag, Adil El-Baouchi, Kamal Hejjaoui, Karthika Rajendran, Abdelaziz Smouni, Fouad Maalouf, Shiv Kumar

**Affiliations:** 1International Center for Agricultural Research in the Dry Areas (ICARDA), Rabat 10112, Morocco; younes.en-nahli@um6p.ma (Y.E.-n.); hasnae_choukri@um5.ac.ma (H.C.); khawlaaloui111@gmail.com (K.A.); 2Laboratoire de Biotechnologie et de Physiologie Végétales, Centre de Recherche BioBio, Faculté des Sciences, Mohammed V University Rabat, Rabat 10112, Morocco; abdelaziz.smouni@um5.ac.ma; 3Materials Science Center, Ecole Normale Supérieure, LPCMIO, Mohammed V University of Rabat, Rabat 10100, Morocco; 4AgroBioSciences Program (AgBS), College of Sustainable Agriculture and Environmental Science (CSAES), University Mohammed VI Polytechnic (UM6P), Ben Guerir 43150, Morocco; adil.elbaouchi@um6p.ma (A.E.-B.); kamal.hejjaoui@um6p.ma (K.H.); 5Laboratory of Ecology and Environment, Ben M’Sick Faculty of Sciences, University Hassan II, Casablanca 20800, Morocco; 6Biotechnology Research Unit, Regional Center of Agricultural Research of Rabat, National Institute of Agricultural Research (INRA), Rabat 10090, Morocco; rachidmentag@yahoo.ca; 7Vellore Institute of Technology (VIT), VIT School of Agricultural Innovations and Advanced Learning (VAIAL), Vellore 632014, India; karthika.rajendran@vit.ac.in; 8International Center for Agricultural Research in the Dry Areas (ICARDA), Beirut 1108 2010, Lebanon; f.maalouf@cgiar.org; 9International Center for Agricultural Research in the Dry Areas (ICARDA), New Delhi 110012, India; sk.agrawal@cgiar.org

**Keywords:** antioxidant activities, catalase, ascorbate peroxidase, superoxide dismutase, proline, flavonoids, sugars, drought stress, heat stress

## Abstract

Climate change has significantly exacerbated the effects of abiotic stresses, particularly high temperatures and drought stresses. This study aims to uncover the mechanisms underlying heat and drought tolerance in lentil accessions. To achieve this objective, twelve accessions were subjected to high-temperature stress (32/20 °C), while seven accessions underwent assessment under drought stress conditions (50% of field capacity) during the reproductive stage. Our findings revealed a significant increase in catalase activity across all accessions under both stress conditions, with ILL7814 and ILL7835 recording the highest accumulations of 10.18 and 9.33 under drought stress, respectively, and 14 µmol H_2_O_2_ mg protein^−1^ min^−1^ under high temperature. Similarly, ascorbate peroxidase significantly increased in all tolerant accessions due to high temperatures, with ILL6359, ILL7835, and ILL8029 accumulating the highest values with up 50 µmol ascorbate mg protein^−1^ min^−1^. In contrast, no significant increase was obtained for all accessions subjected to water stress, although the drought-tolerant accessions accumulated more APX activity (16.59 t to 25.08 µmol ascorbate mg protein^−1^ min^−1^) than the sensitive accessions. The accessions ILL6075, ILL7814, and ILL8029 significantly had the highest superoxide dismutase activity under high temperature, while ILL6363, ILL7814, and ILL7835 accumulated the highest values under drought stress, each with 22 to 25 units mg protein^−1^. Under both stress conditions, ILL7814 and ILL7835 recorded the highest contents in proline (38 to 45 µmol proline/g FW), total flavonoids (0.22 to 0.77 mg QE g^−1^ FW), total phenolics (7.50 to 8.79 mg GAE g^−1^ FW), total tannins (5.07 to 20 µg CE g^−1^ FW), and total antioxidant activity (60 to 70%). Further, ILL7814 and ILL6338 significantly recorded the highest total soluble sugar content under high temperature (71.57 and 74.24 mg g^−1^, respectively), while ILL7835 achieved the maximum concentration (125 mg g^−1^) under drought stress. The accessions ILL8029, ILL6104, and ILL7814 had the highest values of reducing sugar under high temperature with 0.62 to 0.79 mg g^−1^, whereas ILL6075, ILL6363, and ILL6362 accumulated the highest levels of this component under drought stress with 0.54 to 0.66 mg g^−1^. Overall, our findings contribute to a deeper understanding of the metabolomic responses of lentil to drought and heat stresses, serving as a valuable reference for lentil stress tolerance breeding.

## 1. Introduction

Crops are subjected to a wide range of abiotic stresses such as drought, heavy metals, UV rays, heat, and salinity, concurrently throughout their life cycles due to constantly changing environmental conditions, which reduce and limit their growth and productivity [1,2]. Extreme temperatures and drought stress have been identified as two of the most severe environmental risks restricting crop development, productivity, and, consequently, food security in a changing climate [3]. These two stresses contribute significantly to reducing the production of several important staple food crops, which provide 60% of the world’s food energy [4,5,6,7]. Drought stress and high temperatures have been on the rise in recent decades, and this trend is expected to continue and worsen in the future [8]. Consequently, the resulting conditions are likely to have a significant impact, leading to increased malnutrition and micronutrient deficiencies, particularly in regions such as sub-Saharan Africa, South Asia, Central and South America, and small islands.

Lentil (*Lens culinaris* Medik.) holds a significant position as the third most vital cool-season food legume globally, after chickpea and pea [9], with an area of 5.5 million ha, a production of 5.6 million tons, and a productivity of 1004 Kg ha^−1^ in 2021 [10]. The cultivation of lentil predominantly takes place in rainfed environments, where drought and extreme temperatures are the two most challenging abiotic stresses that reduce its production in arid and semi-arid areas [11,12]. Compared to other leguminous crops, lentil has been found to exhibit greater sensitivity to high temperatures and drought stress, particularly during the reproductive stage, significantly diminishing its yield and productivity [13].

The impacts of extreme temperatures and water stress are receiving increased attention due to the substantial threat to the productivity of leguminous crops. These stresses adversely affect crucial factors such as pollen viability, fertilization, and ultimately, the formation of pods [14]. Elevated temperatures exceeding 32/20 °C (max/min) during the flowering and pod-filling stages have a harmful effect on the growth and grain filling of lentil, and by consequence, negatively impact both yield and nutritional quality [15,16]. Extensive research has been conducted on the effects of high temperatures and drought stress across various plant species, including wheat [17], barley [18], maize [19], chickpea [20], lentil [21], and soybean [22].

Both high temperatures and drought stress induce an imbalance between reactive oxygen species (ROS), such as superoxide radical (O^•−^_2_), hydrogen peroxide (H_2_O_2_), hydroxyl radicals (OH•), singlet oxygen (^1^O_2_), and the scavenging activities of antioxidant system components in plants [23]. Abiotic stress causes excessive ROS production, which interferes with photosynthesis, increases lipid peroxidation and electrolyte leakage, damages the structures of cell membranes, and disrupts nucleic acids and proteins. As a result, cells are unable to perform their normal functions [24]. The accumulation of free amino acids, proteins, solutes proline, soluble sugars, and protective heat shock proteins and activation of enzymatic (ascorbate peroxidase, catalase, superoxide dismutase, and peroxidase) and non-enzymatic (glutathione, tocopherols, carotenoids, and ascorbate) antioxidant systems are some of the mechanisms used by plants [25,26]. In addition, secondary metabolites such as phenols, flavonoids, and tannins play a pivotal role in the adaptation and defense of plants against deleterious environmental factors like heat and drought stresses [27]. Accumulation of secondary metabolites in plants during stress conditions contributed to enhance physiological processes and stress tolerance via detoxifying ROS, ultimately promoting plant growth and development [28,29]. Hence, it is important to investigate and understand the enzymatic and non-enzymatic responses to high temperatures and drought stress to improve plants’ stress tolerance.

In order to increase lentil production in areas that are severely impacted by climate change, it is essential to investigate the mechanisms linked to tolerance of high temperatures (>32/20 °C) and drought stress. Despite the importance of enzymatic and non-enzymatic responses in the lentil defense system against the increasing frequency of high temperatures and drought stress, there have been limited studies conducted on this aspect [30,31,32]. In this context, this investigation is the continuation of our previous efforts, and it provides additional insight into the responses of different metabolic traits to drought and heat stresses in lentil accessions. The objectives of the current study were (i) to uncover the metabolic mechanisms underlying heat and drought tolerance in lentil accessions selected from our previous research findings [33,34] and to (ii) identify superior lentil accessions best suited for heat and drought tolerance improvement in lentil germplasm for future programs.

## 2. Results

### 2.1. Antioxidant Enzyme Activity (CAT, APX, and SOD)

Seven accessions were subjected to the drought stress (50% field capacity), while twelve accessions were exposed to high-temperature stress (32/20 °C) during the reproductive phase. Our findings showed that stress significantly increased catalase (CAT) activity to higher than in the control plants, especially in the heat-tolerant accessions (6.91 to 16.32 µmol H_2_O_2_ mg protein^−1^ min^−1^) and drought-tolerant accessions (4.12 to 10.18 µmol H_2_O_2_ mg protein^−1^ min^−1^) compared to the sensitive accessions (Figure 1a,b). The accessions ILL6359, ILL7814, and ILL7835 maintained the highest CAT accumulation with an increase of 56 to 65%; however, ILL5919 and ILL6075 recorded the lowest values under high-temperature conditions with 6.24 and 6.91 µmol H_2_O_2_ mg protein^−1^ min^−1^, respectively. In response to drought stress, the accessions ILL6363, ILL7835, and ILL7814 retained the highest CAT activity (9.32 to 10.18 µmol H_2_O_2_ mg protein^−1^ min^−1^) with an increase of 27 to 47% compared to the control plants. In contrast, our findings showed that drought stress decreased CAT activity in the sensitive accessions ILL5919 and ILL7820 by 21 and 36%.

Under high temperature, ascorbate peroxidase (APX) significantly increased (*p* < 0.05) in all accessions except ILL5919, ILL6075, ILL6104, and ILL6338, which had values statistically similar to the control plants (Figure 1c). The tolerant accessions ILL6359, ILL8029, and ILL7835 were able to retain the highest APX activity (52.55 to 55.21 µmol ascorbate mg protein^−1^ min^−1^) in response to high-temperature stress with a 39 to 69% increase compared to the control. In contrast, no significant increase was obtained for all accessions subjected to water stress, although the drought-tolerant accessions accumulated more APX activity (16.59 t to 25.08 µmol ascorbate mg protein^−1^ min^−1^) than the sensitive accessions. The highest APX activity was recorded for the accessions ILL7835 and ILL7814 with 21.31 and 25.08 µmol ascorbate mg protein^−1^ min^−1^, respectively (Figure 1d). In addition, our findings indicated that APX activity in the accessions ILL5919, ILL7820, ILL6075, and ILL6363 decreased by 46 to 51% as a result of drought stress.

Superoxide dismutase (SOD) activity significantly increased in all accessions except ILL5919 under heat stress and ILL7820 under drought stress when compared to the control (Figure 1e,f). Under high-temperature stress, the accessions ILL6075, ILL6104, ILL7223, ILL7814, and ILL8029 accumulated the highest SOD activity (20 to 25 units mg protein^−1^) with a 39 to 65% increase compared to control plants. However, the drought-tolerant accessions ILL6362, ILL6363, ILL7814, and ILL7835 retained higher SOD activity 21.82 to 27.43 units mg protein^−1^ than sensitive accessions (9.79 to 16.02 units mg protein^−1^) in response to drought stress conditions.

### 2.2. Effect of High-Temperature and Drought Stress on Proline Content

Proline content (PC) significantly increased (*p* < 0.05) due to high-temperature and drought stress in all accessions except the susceptible accessions (ILL5919 and ILL7820), which had statistically similar PC under both stresses in comparison with the control (Figure 2). Under high-temperature stress, the maximum value of PC was found in ILL7814, ILL7833, and ILL7835 (up to 45 µmol proline/g FW with a 55 to 71% increase compared to control plants). However, ILL6362, ILL7814, and ILL7835 had the highest PC with 34.63, 38.09, and 35.78 µmol proline/g FW, respectively, under drought stress conditions.

### 2.3. Total Flavonoid Content Response to High-Temperature and Drought Stress

Total flavonoid content (TFC) significantly increased (*p* < 0.05) in five accessions (ILL6075, ILL6104, ILL6338, ILL7833, and ILL7835) in response to high temperature, whereas all accessions except ILL5919 exposed to drought stress increased TFC compared to normal conditions (Figure 3). Under high-temperature stress conditions, ILL6104, ILL6338, and ILL7835 increased TFC by up 70% and accumulated 0.46, 0.64, and 0.77 mg QE g^−1^ FW, respectively. Under drought stress, accessions ILL6362, ILL6363, and ILL7814 had the highest TFC (up to 0.22 mg QE g^−1^ FW) with an increase of 8 to 60% compared to control plants.

### 2.4. Total Phenolics Content Response to High-Temperature and Drought Stress

When subjected to high-temperature stress, all accessions displayed a significant increase in their total phenolics content (TPC), with the exception of ILL5919, which did not exhibit a significant increase compared to the control (Figure 4a). The TPC accumulation was greatest in ILL6104, ILL6338, ILL7814, ILL7835, and ILL8029 (7.50 to 8.79 mg GAE g^−1^ FW) with a 77 to 90% increase compared to control plants. However, only the two tolerant accessions ILL7814 and ILL7835 significantly increased TPC by up to 85% in response to drought stress (Figure 4b). In drought conditions, the average TPCs of ILL7814 and ILL7835 were 7.67 and 8.79 mg GAE g^−1^ FW, respectively.

### 2.5. Tannin Content Response to High-Temperature and Drought Stress

High-temperature stress caused tannin content (TC) to increase significantly (*p* < 0.05) in comparison to the control in all accessions, while ILL5919 showed no significant difference (Figure 5a). The highest TC accumulation under high-temperature stress was found in accessions ILL6104, ILL7814, ILL7835, and ILL8029 (20.31 to 23.92 µg CE g^−1^ FW), which increased their TC by 80 to 90% compared to control. With drought stress, on the other hand, TC significantly decreased in accessions ILL5919, ILL6075, and ILL6362 by 65 to 78%, while it increased TC in ILL7814 and ILL7835 by up 50% with an average of 5.07 and 8.37 µg CE g^−1^ FW, respectively (Figure 5b).

### 2.6. Total Soluble Sugar and Reducing Sugar under Stressed Conditions

Total soluble sugar (TSS) were significantly augmented by 30 to 66% in eight accessions in response to high-temperature conditions, with the highest accumulation recorded in ILL7814 and ILL6338 with 71.57 and 74.24 mg g^−1^, respectively (Figure 6a). Furthermore, TSS accumulation significantly increased by up 60% in three accessions (ILL5919, ILL6075, and 7835) in response to drought stress (Figure 6b). Among all the accessions, ILL7835 demonstrated the highest TSS content, with an average of 125 mg g^−1^ in response to drought stress conditions.

Compared to normal conditions, reducing sugar (RS) significantly increased (*p* < 0.05) by 40 to 60% in response to high-temperature stress in seven accessions (Figure 6c). The accessions ILL8029, ILL6104, and ILL7814 accumulated the highest RS under high temperature, with 0.62, 0.72, and 0.79 mg g^−1^, respectively. On the other hand, the accessions ILL6075, ILL6363, and ILL7814 significantly increased RS by 30 to 50% under drought stress, whereas ILL5919, ILL6362, ILL7820, and ILL7835 had statistically similar RS compared to normal conditions (Figure 6d). Under drought stress, ILL6075, ILL6363, and ILL6362 had the highest RS with 0.54, 0.56, and 0.66 mg g^−1^, respectively, whereas ILL5919 and ILL7820 had the lowest RS with 0.20 and 0.27 mg g^−1^, respectively.

### 2.7. Total Antioxidant Activity Response to High-Temperature and Drought Stress

Total antioxidant activity (TAA) measurements in all 12 accessions under high-temperature stress showed that seven accessions had significantly (*p* < 0.05) higher TAA than the control. The heat-tolerant accessions ILL6104, ILL6338, ILL7814, ILL7835, and ILL8029 had the top TAA with 60 to 74% of DPPH inhibition in high-temperature conditions (Figure 7). However, TAA of accessions ILL7814, ILL6363, and ILL7835 significantly increased by 50 to 67% in response to drought stress, with 60.05, 62.79, and 65.82% of DPPH inhibition, respectively. Under the influence of both high-temperature and drought stress, the susceptible accessions ILL5919 and ILL7820 displayed the lowest accumulation of TAA.

### 2.8. Genotypic Correlation under Normal, High-Temperature, and Drought-Stress Conditions

In normal conditions, TFC exhibited a highly significant positive correlation (*p* < 0.01) with TC and TPC, while it was negatively correlated at *p* < 0.05 with SOD (r = −0.57) (Figure 8a). Furthermore, a remarkably significant positive correlation (*p* < 0.001) between TPC and TC was found (r = 0.88). The heatmap analysis showed that CAT, APX, SOD, and RS were grouped together in the same cluster, while TSS and PC were found in the second cluster (Figure 8b). Under normal conditions, TC, TPC, TFC, and TAA are grouped together. Additionally, genotypic correlation identified three groups. The first group included two heat-tolerant accessions (ILL8029 and ILL6104), two heat- and drought-tolerant accessions (ILL7814 and ILL7835), and one susceptible accession (ILL7820). The second group consisted of two moderately heat- and drought-tolerant accessions (ILL7223 and ILL8025), one drought-tolerant accession (ILL6075), one heat-tolerant accession (ILL7833), and one susceptible accession ILL5919. Three accessions were grouped in one group: two moderately tolerant (ILL6363 and ILL6359), one drought-tolerant (ILL6362), and one heat-tolerant (ILL6338).

In high-temperature conditions, SOD displayed a notable positive correlation with TAA, TC, PC, and RS, whereas CAT had positive correlation (*p* < 0.01) with APX (r = 0.72) (Figure 9a). Moreover, TPC, TAA, and TC were found to have a highly positive correlation (*p* < 0.001). The heatmap analysis identified that CAT and APX were clustered together, while TFC was clustered independently (Figure 9b). The remaining components formed the third group. Three heat-tolerant accessions (ILL8029, ILL6104, and ILL6338) and two accessions tolerant to high-temperature and drought stress (ILL7814 and ILL7835) were clustered together in the same group based on genotypic correlation. For almost all the tested components, these accessions had the highest accumulations. The second group contained two moderately tolerant (ILL7223 and ILL8025), one heat-tolerant (ILL7833), one drought-tolerant (ILL6075), and one sensitive (ILL7820) accession. The third group included one susceptible accession (ILL5919) and one moderately tolerant accession (ILL6359).

Under drought stress conditions, SOD was positively correlated with CAT, TAA, and TC (Figure 9c). In addition, APX was found to be positively related to CAT, PC (*p* < 0.05), and TPC (*p* < 0.01). Furthermore, TAA exhibited a positive correlation with TC at *p* < 0.001, whereas PC had a positive correlation with RS (r = 0.83) under drought stress. The heatmap analysis revealed that CAT, SOD, APX, TPC, TAA, and TC were clustered in the same group, whereas RS, PC, and TFC formed the second group (Figure 9d). In the third group, TSS was clustered individually. Furthermore, accessions ILL7835, ILL6363, and ILL7814 were found in the same cluster, while susceptible accessions ILL7820 and ILL5919 were found in a separate group. The accessions ILL6075 and ILL6362, which are drought-tolerant, were found in the third group.

## 3. Discussion

In the near future, alterations in climatic conditions are anticipated to lead to a frequent occurrences of droughts and extreme temperatures, thereby posing a substantial threat to food security [35]. Both stress factors impose a detrimental effect on metabolic processes, affecting vital aspects like photosynthesis, water relations, nutrient absorption, and consequently, grain yield [36]. Hence, the selection of tolerant genotypes to heat and drought stresses and the comprehension of the underlying mechanisms by plant breeding scientists are extremely important for increasing agricultural productivity in arid and semi-arid environments. In the current study, fourteen lentil accessions were subjected to high-temperature stress (32/20 °C) throughout the critical reproductive stage, while seven accessions were assessed under drought stress by reducing field capacity to 50%. The two sets of accessions were chosen based on their tolerance to high temperatures and drought stress discovered from our previous investigations [33,34]. Under both stressed conditions, the accessions ILL5919 and ILL7820 were used as susceptible controls. The outcomes of our study revealed that the application of high-temperature and drought-stress conditions induced substantial modifications in both enzymatic and biochemical processes in the tested lentil accessions. Remarkably, the concentrations of ascorbate peroxidase, catalase, and superoxide dismutase exhibited a significant increase in the heat-tolerant accessions as compared to the heat-sensitive ones. This suggests a stronger enzymatic response to high-temperature stress in the heat-tolerant accessions. Tolerant accessions ILL7814 and ILL7835 significantly increased CAT and SOD activity in response to heat and drought stresses; however, there was no significant difference in APX across all accessions exposed to drought stress. The moderately tolerant accession ILL6363 accumulated the highest SOD under drought stress, whereas the tolerant accessions ILL6075 and ILL6362 had significant SOD activity. Additionally, it was observed that under both high-temperature and drought-stress conditions, the susceptible accessions ILL5919 and ILL7820 exhibited the lowest antioxidant activity, which suggests their reduced capacity to combat oxidative stress under these challenging conditions. When subjected to high-temperature stress, the moderately heat-tolerant accessions (ILL6075, ILL6359, ILL7223, and ILL8025) displayed a significant increase in the levels of CAT, APX, and SOD enzymes compared to normal conditions. SOD activity was also significantly increased in the ILL6363 accession, which is moderately drought-tolerant.

Both drought and high-temperature stresses can result in elevated production of reactive oxygen species, which hinder various metabolic processes in plants, including those related to photosynthetic components. As a protective mechanism, plants deploy antioxidant enzymes such as CAT, APX, and SOD as their primary defense to scavenge H_2_O_2_ with different mechanisms and suppress its toxic impacts [37]. Our study observed heightened activities of antioxidant enzymes in nearly all tolerant and moderately tolerant lentil accessions, as a response to the stresses of high temperature and drought, which was in agreement with the outcomes of other investigations [38,39,40,41,42]. Under adverse environmental conditions, complex antioxidant enzyme activities play a critical role in overcoming uncontrolled reactive oxygen species generation and protecting plants from oxidative damage [23]. Through our research, we discovered significant variations in CAT, APX, and SOD activities between tolerant and sensitive accessions. In fact, the tolerant accessions exhibited higher levels of antioxidant enzyme activities compared to the susceptible ones. In lentil, Chakraborty and Pradhan [43] also observed an elevation in APX, SOD, and CAT levels in response to high-temperature stress, with the tolerant accessions demonstrating a more pronounced increase. However, our findings indicate that APX activity did not exhibit a significant increase in lentil accessions under drought stress. Furthermore, the APX activity for four specific accessions (ILL5919, ILL7820, ILL6363, and ILL6075) displayed a non-significant decrease, suggesting a lower level of oxidative stress due to induced antioxidant enzyme activities and detoxified reactive oxygen species, as elucidated by Singh et al. [41]. Reduction in CAT and APX activities was also reported in two grasses and two legume species that were subjected to high temperatures and drought stress [44]. Nonetheless, in comparison to the sensitive controls, the majority of drought-tolerant accessions exhibited notably higher levels of APX and CAT. Similar results regarding antioxidant enzyme activities were reported in other studies in response to heat and drought stresses [45,46]. In addition, our findings confirmed the results of Sarker and Oba [42] in demonstrating that accessions with higher tolerance to drought stress had higher SOD activity contribution compared to accessions with lower tolerance to water stress. Therefore, these enzymatic antioxidants (CAT, APX, and SOD) can be used as economical and effective metabolic indicators for either screening or enhancing lentil germplasm for heat and drought tolerance in the early growth stages.

In lentil, Hosseini et al. [47] identified two groups of SOD genes (Cu/Zn-SOD and Mn-SOD) up-regulated in mitochondria, chloroplast, and cytosol under drought- and heat-stress conditions. Furthermore, previous investigations also showed that SOD activity increased in drought-tolerant genotypes of lentil under drought stress [48], and chloroplastic and cytosolic Cu/Zn-SOD and Mn-SOD were up-regulated under drought stress in soybean [49], wheat [50], and rice [51]. In addition, several cytosolic and chloroplastic APX genes such as lentil homolog of Atperoxidase 1, 12, 52, 67, and 17 were commonly up-regulated under drought and heat stress, and they were found to be important genes used as indicators of tolerance to heat and drought-stress conditions [47]. Rahman et al. [52] also reported that the expressions of APX, Cu/Zn-SOD, and CAT genes are the key candidates in antioxidant defense, which might be useful in molecular breeding strategies for heat and drought tolerance.

Proline stands as one of the most crucial osmoprotectants in plants, serving as a solute that adapts to alterations in the cell’s water environment and mitigating the adverse impacts of high temperatures and drought stress on plants [53]. After the removal of stress relief, proline generated during stressful conditions can additionally function as a reservoir for energy and ammonia sources directly influencing plant metabolism [4]. Therefore, higher accumulations of proline are associated with increased stress tolerance. Proline also acts as an antioxidative defense molecule, and it scavenges ROS to activate specific gene functions that are essential for the plant’s recovery from stresses [54]. In the current investigation, we observed a significant increase in proline content in all tested accessions subjected to high-temperature and drought stresses, except for the susceptible accessions ILL5919 and ILL7820, which displayed non-significant increases under these stress conditions. These findings align with the observations of Singh et al. [41] who reported higher proline levels in heat-tolerant lentil accessions when compared to sensitive ones. Increasing proline content was also reported by Hosseini et al. [47] in response to heat and drought stresses, and it was identified the hub gene P5CS2 (delta 1-pyrroline-5-carboxylate synthase 2) as the major transcription factor involved in lentil tolerance to these stresses. Furthermore, Hayat et al. [55] highlighted the role of proline in preventing photodamage to chloroplast thylakoid membranes by effectively scavenging and/or reducing the production of reactive oxygen species. Moreover, the utilization of exogenous proline has been shown to regulate drought stress by stimulating plant growth, which is achieved by triggering the antioxidant mechanism, reducing oxidative damage, enhancing the synthesis of compatible solutes, and accelerating proline accumulation, resulting in enhanced photosynthesis and yield attributes [56]. El-Beltagi et al. [57] found that the sequence combination of antioxidant and proline increased the antioxidant defense system and osmolyte synthesis, resulting in accelerated plant growth in chickpea under drought stress. Exogenous proline has the remarkable capacity to interact with enzymes, effectively preserving protein aggregation and preventing thermal denaturation [58].

The soluble sugar and reducing sugar are also among the most important osmotic regulators in plants, serving diverse cellular functions including energy storage, osmoprotectants, and signals for adapting to abiotic stresses [59]. Under the influence of high-temperature and drought stress, the concentrations of total soluble sugar increased in almost all examined accessions within our study. The same trend was also reported by Shah et al. [31] who observed a significant rise in soluble sugar content and proline content in lentil cultivars exposed to water-deficit stress conditions. Furthermore, Sehgal et al. [60] reported similar outcomes, demonstrating a significant increase in soluble sugar in response to heat and drought stresses in lentil. These results confirmed that heat and drought stresses increased the soluble sugar content in plants and thus reduced the osmotic potential and maintained the normal water demand, as described by Li et al. [61]. On the other hand, reducing sugar content was reported to be correlated with grain yield under heat and drought stresses [32]. Sita et al. [62] found that heat-tolerant lentil accessions accumulated more reducing sugar than sensitive ones, which they attributed to an increase in acid invertase activity (hydrolysis of sucrose into fructose and glucose) in the tolerant accessions, which sustained their redox ionic homeostasis.

In addition, our results revealed that tolerant accessions exposed higher concentrations of total flavonoids, total phenolics, and total tannins than sensitive accessions under both high-temperature and drought-stress conditions. A drastic difference has been also observed in the concentrations of phenols, flavonoids, and tannins of control and drought-stress soybean cultivars [49]. These secondary metabolites function as osmoprotectants and play a crucial role in scavenging free radicals against oxidative damage caused by abiotic stress [63]. According to Hassan et al. [64], total phenolic and flavonoid contents are strong antioxidants, and their accumulation in plants has a strong association with drought-stress tolerance. Ghazghazi et al. [54] concluded that polyphenolic compounds serve as subtracts for the synthesis of enzymes in the antioxidant defense network. Further, the second hypothesis suggests that plants use antioxidant enzymes as their primary method of reducing ROS levels rather than phenolic compounds [65]. Total antioxidant activity significantly increased due to both high-temperature and drought stress. Under stressful conditions, tolerant accessions accumulated a greater total antioxidant concentration than susceptible ones. The heat tolerant accessions ILL6104, ILL6338, ILL7814, and ILL8029 had the highest accumulations of total antioxidant activity in high-temperature conditions, while ILL6363, ILL7814, and ILL7835 were the top accumulators under drought stress, inhibiting DPPH by up to 60%. In the presence of high-temperature and drought stress, the increased antioxidant capacity in tolerant accessions can help mitigate oxidative damage to membranes and proteins, thereby preserving the functionality of organelles and promoting overall plant performance [66].

Based on our findings, proline had a significant positive correlation with CAT and APX under drought stress, and with SOD and TAA under high temperatures. These results may be related to the adequate actions of these antioxidants, which can remove ROS and reduce the harmful effects of stress conditions. These outcomes are consistent with the findings of previous reports in lentil [41,67] and other species [38,68,69]. In addition, we observed a significant correlation between TPC and TAA under high-temperature stress conditions. Similar findings were found in several studies [70,71], suggesting that the total antioxidant activity largely depends on the presence of large amounts of phenolic compounds [72]. Our results also revealed a positive correlation between APX and both CAT and TPC under drought stress. Furthermore, TPC accumulation in plants has been shown to have a strong correlation with their resistance to stress [4]. Similarly, TC was positively correlated with TAA, CAT, and SOD under drought stress and with TAA, SOD, PCA, and PC under conditions of heat stress. Total tannin content in plants has been shown to have significant antioxidant activity for removing and preventing ROS formation [73]. Reducing sugar also had a positive correlation with SOD at high-temperature stress and proline content during drought stress. Hussain et al. [4] found that reducing sugar played a significant role in osmoregulation, which protects plants from environmental conditions.

## 4. Materials and Methods

### 4.1. Plant Material

A set of 14 diverse lentil accessions were selected based on findings from our previous studies [33,34]. The material included four heat-tolerant accessions, two drought-tolerant accessions, two heat- and drought-tolerant accessions, four moderately heat- and drought-tolerant accessions, and two accessions sensitive to both stresses. The material was obtained from the International Center for Agricultural Research in the Dry Areas (ICARDA). The information regarding these accessions can be found in Table 1.

### 4.2. Experiments Details

The experiment was conducted in a controlled glasshouse at ICARDA, Rabat, Morocco (latitude 33°58′45.7″ N longitude 6°51′44.9″ W, and altitude 52 m). In plastic pots (20 cm in diameter and 18 cm in height) filled with compost garden soil and sandy-loam soil in a ratio of 2:2 *w*/*w*, seeds were planted, and each filled pot weighed a total of 3 Kg. In each pot, five seeds were initially sown at a depth of 2 cm and were thinned to three per pot after emergence. Three treatments were used: control (C), drought stress (D), and heat stress (H). Randomized complete block design with four replications was used for each treatment. Plants were kept in normal conditions, which included 25/18 °C (day/night) temperature, 90/60% relative humidity (RH), and 450 μmol m^−2^ s^−1^ light intensity. During development, plants were fully irrigated daily between 09:00 and 10:00 h; however, irrigation was restricted during the flower initiation stage for drought treatment to impose water stress. At the first day of anthesis phase (i.e., first flower appearance), plants in heat treatment were exposed to temperatures above 32 °C for four hours in a growth chamber at 65–70% RH for a period of seven days. By maintaining 50% of the relative water content of the control conditions for nine days, plants under drought stress were subjected to severe water stress. Whole plants were collected in liquid nitrogen and kept at −80 °C for various analyses after being subjected to high temperatures and drought treatments during the reproductive period. Concurrently, plants grown under controlled conditions were also collected.

### 4.3. Total Phenolic Content

Total phenolic content was measured using the technique outlined by Waterhouse [74]. In a 2 mL Eppendorf tube, 200 µL of extract from whole plant and 1 mL of Folin–Ciocalteu reagent were added. After five minutes of incubation, 800 µL of aqueous sodium carbonate 20% was added to the mixture. The mixture was incubated in the dark at room temperature for 60 min. At 765 nm, the absorbance was measured with a Thermo scientific spectrophotometer. The total phenolic concentration was calculated using a gallic acid standard curve. The results were expressed as mg gallic acid equivalents (GAE) per gram of fresh weight. Total phenolic content (TPC) was calculated as follows [33]:(1)$$({TPC)}_{\left(mg\frac{GAE}{g}dw\right)}=(C\times V)/m$$
where C is the total phenolic concentration (mg L^−1^), V is the volume of the extraction, and m is the fresh weight of the plant material (g).

### 4.4. Estimation of DPPH-Scavenging Activity

The antioxidant potential of the extracts from whole plants was determined by their ability to scavenge the stable 1,1-diphenyl-2-picrylhydrazyl (DPPH) free radical, according to Williams et al. [75]. Briefly, 200 µL of each extract preparation was added to 1.8 mL of 0.1 mM DPPH dissolved in methanol. The absorbance of the resulting solution was measured at 517 nm after the agitation of the mixture for 30 min at room temperature. A decrease in absorbance indicates an increase in DPPH-scavenging activity. The scavenging effect was characterized using the following equation:(2)$$scavenging\;activity=\left(\frac{Blank\;absorbance-Sample\;absorbance}{Blank\;absorbance}\right)\times 100$$

### 4.5. Total Soluble Sugar Content

Total soluble sugar concentration from whole plants of each accession was measured using the colorimetric method developed by Dubois et al. [76]. An amount of 2 mL aliquot of a carbohydrate solution was mixed with 1 mL of 5% aqueous solution of phenol in test tubes. Furthermore, a total of 5 mL of concentrated sulfuric acid was rapidly added to the mixture. After allowing the test tubes to stand for 10 min, they were vortexed for 30 s and placed in a water bath at room temperature for 20 min to develop the color. Then, a spectrophotometer was used to record the light absorption at 490 nm. Reference solutions were prepared using the same procedure as described above, except that the 2 mL aliquot of carbohydrate was replaced with bi-distilled water. The phenol used in this procedure was redistilled, and a solution of 5% phenol in water (*w/w*) was prepared immediately before the measurements.

### 4.6. Reducing Sugar Content

The analysis of residual reducing sugar from whole plants of each accession were analyzed using the 3,5-dinitro-salicylic acid (DNS) method described by Miller [77]. In a 25 mL test tube, 1 mL of standard or test sample was added, followed by 3 mL of DNS reagent. The reaction mixture was heated in boiling water for 5 min. The absorbance spectra of the test samples were measured at 540 nm using 2 mm path length quartz cuvettes. The absorbance measurements were compared to a standard mannose curve (180.2 mw) ranging from 0 to 20 mM to identify the presence of reducing sugar released in each extract.

### 4.7. Total Flavonoid Content

Total flavonoid constituents of whole plant extracts were determined using an aluminum chloride colorimetric assay and Quercetine as the standard [78]. An amount of 1 mL of extract was added to a 10 mL flask containing 4 mL of distilled deionized water. A reagent blank using double-distilled H_2_O (ddH_2_O) instead of the extract was prepared at this stage. After adding 0.3 mL of 5% NaNO_2_ to the flask, it was vortexed and kept for five minutes. Then, the mixture was treated with 0.3 mL of 10% AlCl3 solution and left for five minutes. At the sixth minute, 2 mL of 1 M NaOH was added, and the total volume was diluted to 10 mL with ddH_2_O. The solution was vigorously vortexed, and absorbance was measured at 510 nm against the blank using a spectrophotometer. The procedure was repeated with all standard solutions of Quercetine (20–100 mg/L, Sigma-Aldrich, St. Louis, MO, USA) to obtain a standard curve. The total flavonoid content of the extracts was determined and expressed as milligrams of Quercetine equivalents (QE) per gram of extract. All samples were examined in triplicate.

### 4.8. Total Tannin Content

Total tannin content was calculated using the vanillin/HCl method developed by Price et al. [79]. The reagent was prepared by mixing equal parts of methanol solutions of 8% HCl and 1% vanillin, just before the reaction. Aliquots of 0.1 mL of lentil leaf extracts and 0.2 mL of vanillin/HCl reagent were mixed and incubated at 30 °C for 20 min. The absorbances were measured at 500 nm. A standard curve was created by using Catechin methanol solutions ranging from 0.1–2.5 mg mL^−1^. The total tannin content of the extracts was determined and expressed as micrograms of Catechin equivalent (CE) per gram of extracts.

### 4.9. Total Soluble Protein

The method developed by Lowry [80] was utilized to predict the concentration of soluble proteins. An amount of 100 mg from the whole plant was macerated in 0.1 M phosphate buffer (pH = 7.0) and centrifuged at 513.16× *g* for 15 min to obtain a supernatant. The supernatant was treated with 5 mL of TCA (trichloroacetic acid; 15%) and kept at 4 °C for 24 h. The mixture was then centrifuged at 513.16× *g* for 15 min to separate the precipitates. The supernatant was discarded, and the precipitate was dissolved in 0.1 N NaOH (1 mL), allowed to dissolve completely for 18 h, and then treated as an extract.

### 4.10. Proline Content

The proline content was extracted and quantified using the methodology described by Bates et al. [81]. A total of 100 mg of leaf was homogenized in 10 mL of 3% sulphosalicyclic acid and centrifuged at 3000 rpm for 10 min. Filtrate was used for estimation by adding 2 mL of acid ninhydrin and 2 mL of glacial acetic acid. After one hour in a 100 °C water bath, the mixture was transferred to a separating funnel. Then, 4 mL of toluene was added. Two distinct layers were formed, and the absorbance of the upper layer was measured at 520 nm using the toluene as the blank reference.

### 4.11. Antioxidant Enzyme Activity Assays

The activity of catalase (CAT) was determined following the method of Chance and Maehly [82]. Whole plant was ground in mortars using liquid nitrogen. The enzyme was extracted from 0.5 g of tissue using chilled 50 mM sodium phosphate buffer (pH 7.0), containing 1% (*w*/*v*) polyvinylpyrrolidone (PVP), and was centrifuged at 12,000 rpm for 20 min at 4 °C. Enzyme estimation was performed using supernatant. The reaction mixture contained 1.9 mL of 50 mM sodium phosphate buffer (pH 7.5), 0.1 mL enzyme extract, and 1 mL of hydrogen peroxide. The decrease in absorbance after every 30 s interval for 2 min was recorded at 240 nm against a blank. Enzyme activity was calculated using the extinction coefficient of H_2_O_2_ (0.036 mM^−1^ cm^−1^) and was expressed as μmol mg protein^−1^ min^−1^.

Ascorbate peroxidase (APX) activity was measured using the method of Nakano and Asada [83]. A total of 0.5 g of powdered sample was extracted with 2 mL of 50 mM sodium phosphate buffer (pH 7.5) containing 1% PVP and 2 mM ascorbate. The homogenized material was centrifuged at 12,000× *g* for 20 min at 4 °C. Supernatants were used as the crude enzyme source for the enzymatic assays. The reaction mixture contained 50 mM potassium phosphate (pH 7.0), 0.2 mM EDTA, 0.5 mM ascorbic acid, 2% H_2_O_2_, and 0.1 mL enzyme extract in a final volume of 3 mL. The decrease in absorbance at 290 nm for 1 min was recorded, and the amount of ascorbate oxidized was calculated using the extinction coefficient (e = 2.8 mM^−1^ APX was defined as 1 mmol mL^−1^ per min at 25 °C, cm^−1^). Enzyme activity was expressed as μmol mg protein^−1^ min^−1^.

Superoxide dismutase (SOD) activity was assayed by monitoring its ability to inhibit the photochemical reduction of nitro-blue tetrazolium chloride (NBT). One unit of SOD is defined as the amount of enzyme required to inhibit the rate of NBT reduction by 50 percent at 560 nm. In 3 mL of reaction mixture, 75 μM NBT, 1.5 mM riboflavin, 50 mM sodium bicarbonate, 13 mM methionine, 0.1 mM EDTA, 50 mM potassium phosphate buffer (pH 7.5), and 100 μL of enzyme extraction were present. The samples containing the mixture were shaken for 15 min under light at 78 μmol photons s^−1^ m^−1^, and the absorbance at 560 nm was measured. A control reaction mixture that did not exhibit any color development was used as a reference, and its absorbance at 560 nm was subtracted from the absorbance of the reaction solution [84].

### 4.12. Statistical Analysis

A least significant difference (LSD) test was applied at a significance level of *p* < 0.05 to compare the mean values and determine significant differences. Pearson’s correlation coefficient was calculated for the three treatments using metan package in R version 4.1.3 and RStudio version 1.3.31093 [85]. A dendrogram along with a heatmap was performed with the pheatmap R package (Version 1) [86].

## 5. Conclusions

The current study concluded that tolerant lentil accessions exhibited superior responses to high-temperature and drought-stress conditions than sensitive accessions and produced high enzymatic activities for almost all components. These results align with the classification of these accessions as either tolerant or sensitive to high-temperature or drought stress. In addition, we also observed a moderate response of moderately tolerant accessions under both stresses. The inclusion of differentially sensitive accessions allowed us to delve into the mechanisms associated with tolerance to high-temperature and drought stress, which was one of our primary objectives. Under stress, tolerant accessions maintained higher antioxidant activity, osmolytes, and phenolics metabolism. Consequently, these accessions hold promising potential for maintaining essential physiological processes and achieving high yield levels in regions that are significantly affected by increasing temperatures and drought events. However, more research is needed to investigate the underlying mechanism and identify genes associated with heat or drought tolerance in lentil.

## Figures and Tables

**Figure 1 plants-12-03962-f001:**
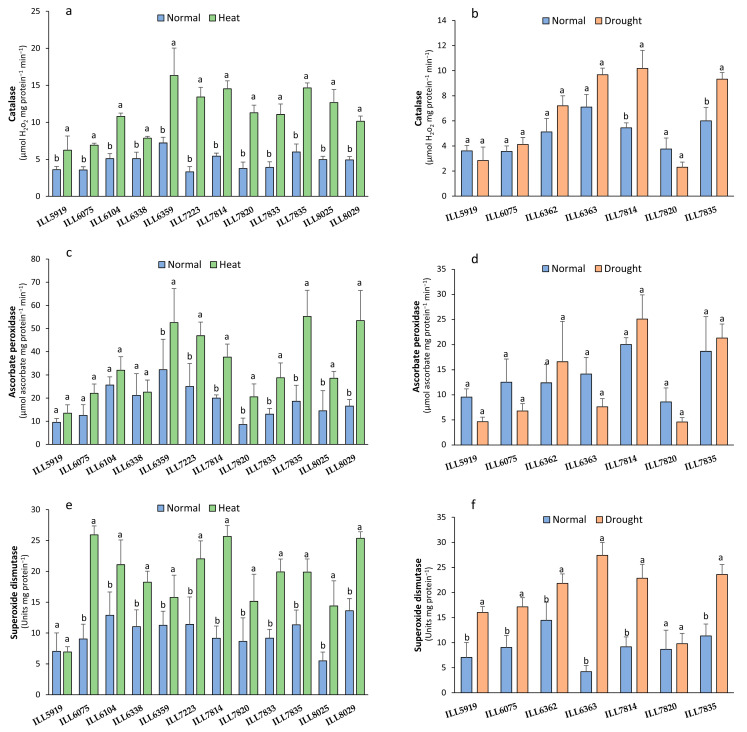
Catalase (CAT), ascorbate peroxidase (APX), and superoxide dismutase (SOD) activities of lentil accessions in response to high-temperature (**a**,**c**,**e**) and drought stress (**b**,**d**,**f**). Significant differences between means of tested accessions under not-stressed and stressed conditions using the LSD test at *p* < 0.05 are indicated by different letters on the bars.

**Figure 2 plants-12-03962-f002:**
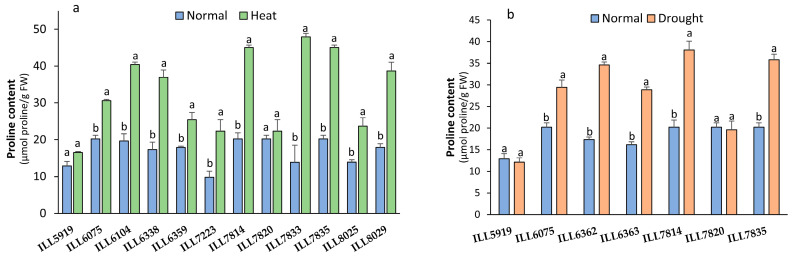
Proline content (PC) response to heat stress (**a**) and drought stress (**b**) in lentil accessions. Significant differences between means of tested accessions under not-stressed and stressed conditions using the LSD test at *p* < 0.05 are indicated by different letters on the bars.

**Figure 3 plants-12-03962-f003:**
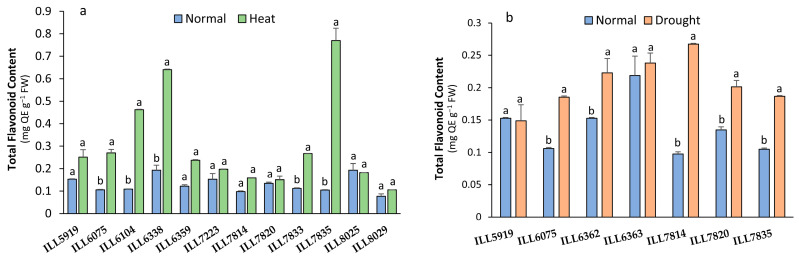
Total flavonoid content (TFC) of lentil accessions in response to heat stress (**a**) and drought stress (**b**). Significant differences between means of tested accessions under not-stressed and stressed conditions using the LSD test at *p* < 0.05 are indicated by different letters on the bars.

**Figure 4 plants-12-03962-f004:**
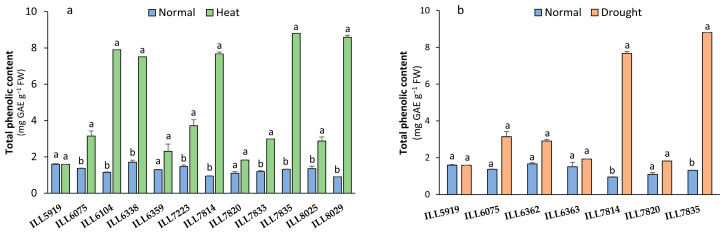
Total phenolic content (TPC) of lentil accessions in response to heat stress (**a**) and drought stress (**b**). Significant differences between means of tested accessions under not-stressed and stressed conditions using the LSD test at *p* < 0.05 are indicated by different letters on the bars.

**Figure 5 plants-12-03962-f005:**
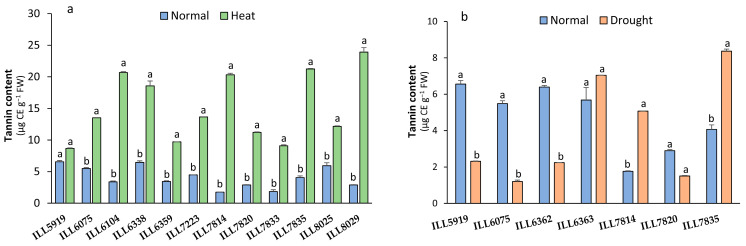
Tannin content (TC) in response to heat stress (**a**) and drought stress (**b**) in lentil accessions. Significant differences between means of tested accessions under not-stressed and stressed conditions using the LSD test at *p* < 0.05 are indicated by different letters on the bars.

**Figure 6 plants-12-03962-f006:**
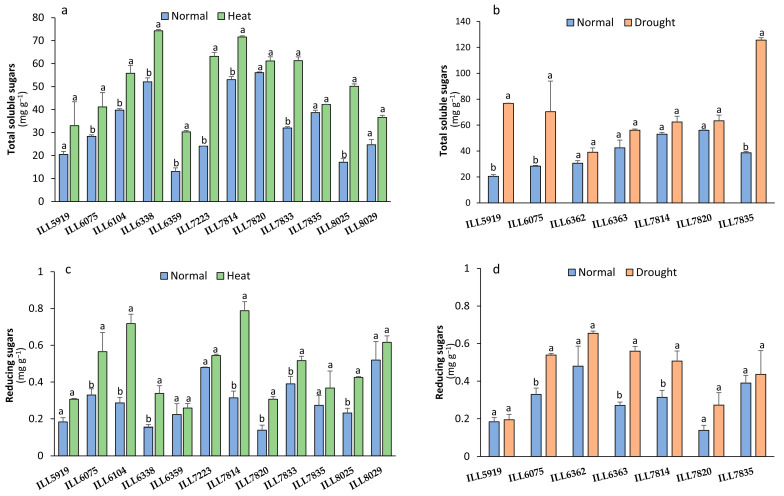
Total soluble sugar (TSS) and reducing sugar (RS) in response to heat stress (**a**,**c**) and drought stress (**b**,**d**) in lentil accessions. Significant differences between means of tested accessions under not-stressed and stressed conditions using the LSD test at *p* < 0.05 are indicated by different letters on the bars. The colored bars represent the treatments as blue for non-stress conditions, green for high temperatures, and orange for drought stress.

**Figure 7 plants-12-03962-f007:**
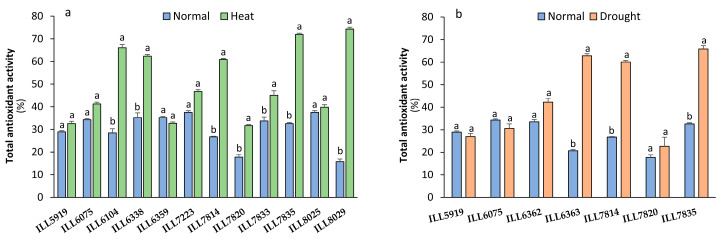
Total antioxidant activity (TAA) in lentil accessions expressed as a percentage of DPPH inhibition in response to heat stress (**a**) and drought stress (**b**). Significant differences between means of tested accessions under not-stressed and stressed conditions using the LSD test at *p* < 0.05 are indicated by different letters on the bars.

**Figure 8 plants-12-03962-f008:**
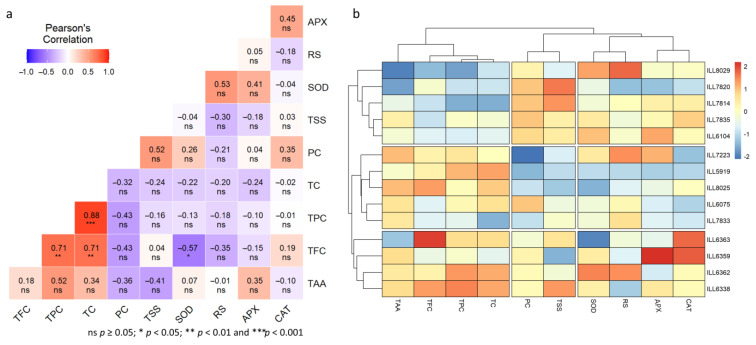
Pearson’s correlation coefficients (**a**) and dendrogram along with heatmap (**b**) between various traits of lentil accessions under normal conditions.

**Figure 9 plants-12-03962-f009:**
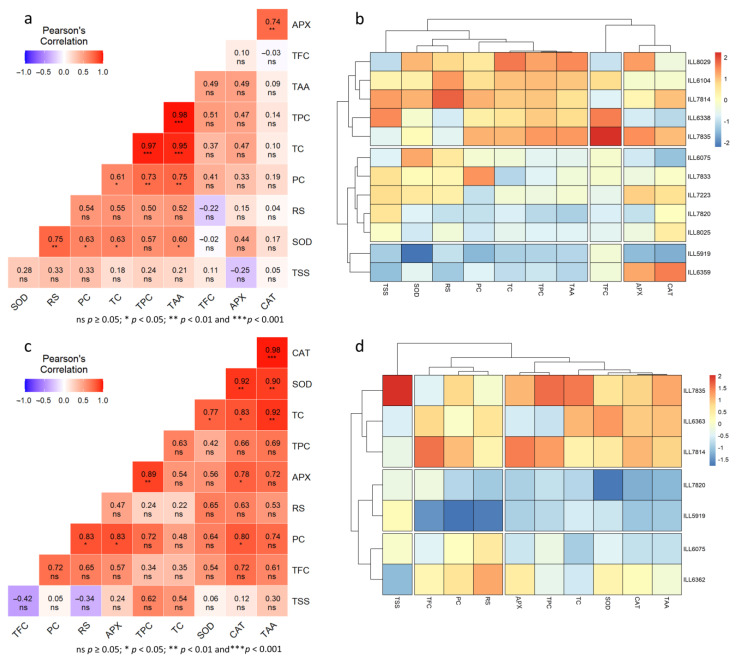
Pearson’s correlation coefficients (**left**) and dendrogram along with heatmap (**right**) between various traits for high-temperature (**a**,**b**) and drought-stress (**c**,**d**) conditions.

**Table 1 plants-12-03962-t001:** Description of origin, IG number, and classification of lentil accessions used in the current study.

Accession	IG Number	Origin	Classification
ILL5919	69528	Ethiopia	HDS
ILL6075	70130	Pakistan	DT
ILL6104	70159	Pakistan	HT
ILL6338	71270	Pakistan	HT
ILL6359	71291	Pakistan	MT
ILL6362	71294	Pakistan	DT
ILL6363	71295	Pakistan	MT
ILL7223	75942	Nepal	MT
ILL7814	114931	Nepal	HDT
ILL7820	114951	Nepal	HDS
ILL7833	115006	Nepal	HT
ILL7835	115010	Nepal	HDT
ILL8025	117726	Pakistan	MT
ILL8029	117734	Pakistan	HT

HT, heat-tolerant; DT, drought-tolerant; MT, moderately tolerant to heat and drought; HDT, tolerant to heat and drought; HDS, susceptible to heat and drought; IG, ICARDA germplasm.

## Data Availability

Datasets generated and/or analyzed during the current study are available from the corresponding author upon reasonable request.

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
