# Peer review of "Metabolic Mechanisms Underlying Heat and Drought Tolerance in Lentil Accessions: Implications for Stress Tolerance Breeding"

_plants, 2023, doi:10.3390/plants12233962_

Round 1

Reviewer 1 Report

1.     Please revise the title. Potential suggestion “Metabolic Mechanisms Underlying Heat and Drought Tolerance in Lentil Accessions: Implications for Stress Tolerance Breeding”

Or

Metabolic Responses at the Reproductive Stage: Unveiling Heat and Drought Tolerance Mechanisms in Lentil Accessions for Stress Tolerance Breeding

2.     Abstract should be improved to have more attractiveness, particularly, about the key findings. It needs a more scientific approach rather than supplying generic information.

3.     I would recommend revisiting the research objectives just to enhance the study's direction. I would also recommend investing additional effort into the conclusions to maximize the paper's overall impact. 

4.     Make the legends of the figures as descriptive as possible and make sure to add information about the figure in an easy-to-understand way.

5.     Improve the figure/picture quality.

6.     Revise the keywords of the article.

7.     Please check the formatting and editing according to the journal's requirements. This also includes the reference and bibliography.

8.     Please cite more relevant literature in the manuscript from recent years (2022-2023).

9.     Please mention the reason to select Seven accessions for drought stress and twelve accessions for high-temperature stress.

10.  Please include pictures of plants during stress treatment or controlled conditions.

11.  Please include genetic references for the role of biochemical and enzymatic features in stress tolerance in lentils or other plants

12.  Please mention the identities of better-performing accessions in the abstract.

Line 47: Define the “range of abiotic stresses” to enhance clarity.

Line 49: “damage to agricultural crop growth” can be rewritten for better understanding.

Line 52-53: Rephrase the line to enhance clarity.

Line 59: Capitalize the first letter of the words at the beginning of a sentence.

Line 59: Replace "holds the significant position of being" with "holds a significant position as the".

Line 62: Rewrite the sentence to provide a better understanding of the concept to the reader.

Line 66: Change "diminishing its production and productivity" to "diminishing its yield and productivity".

Line 68: Correct "receiving increased attention due to their substantial threat" to "receiving increased attention due to the substantial threat".

Line 94: What do you mean by “high temperature”? can you mention the numbers?

Line 101: high temperatures stress (32/20 °C)

Line 394: You mentioned the “anthesis phase” Kindly define it when using it for the first time. 

Quality of English Language is good, however minor improvements can be made. 

Author Response

To: Ms. Marijana Apic

Journal Name: Plants (ISSN 2223-7747)

Special Issue: Abiotic Stress-Induced Secondary Metabolites Regulating Plant Metabolism

Section: Plant Physiology and Metabolism

Date: August 17, 2023

Revision for the Manuscript ID: plants-2557895

Dear Editor,

We appreciate you and the reviewers for your precious time in reviewing our paper and providing valuable comments. We have carefully considered the comments and tried our best to address every one of them. We hope the manuscript after careful revisions meet your high standards. The authors welcome further constructive comments if any.

Below we provide the point-by-point responses to the comments of the reviewer 1 alongside this letter. Track changes was used to highlight the changes in the manuscript, these changes have been approved by all co-authors.

Sincerely,

Dr. EL HADDAD Noureddine

Point 1:     Please revise the title. Potential suggestion “Metabolic Mechanisms Underlying Heat and Drought Tolerance in Lentil Accessions: Implications for Stress Tolerance Breeding”

Or

Metabolic Responses at the Reproductive Stage: Unveiling Heat and Drought Tolerance Mechanisms in Lentil Accessions for Stress Tolerance Breeding

Response 1: Thank you for your suggestion.  We have used the first proposition “Metabolic Mechanisms Underlying Heat and Drought Tolerance in Lentil Accessions: Implications for Stress Tolerance Breeding”.

Point 2:     Abstract should be improved to have more attractiveness, particularly, about the key findings. It needs a more scientific approach rather than supplying generic information.

Response 2: The abstract was improved, and more information about our key findings was added.

Point 3:     I would recommend revisiting the research objectives just to enhance the study's direction. I would also recommend investing additional effort into the conclusions to maximize the paper's overall impact. 

Response 3: The objectives were revised and we made more efforts to improve the conclusions of our manuscript.

Point 4:     Make the legends of the figures as descriptive as possible and make sure to add information about the figure in an easy-to-understand way.

Response 4: Legends were added on figures and more information was included.

Point 5:     Improve the figure/picture quality.

Response 5: The quality of all figures were improved.

Point 6:     Revise the keywords of the article.

Response 6: Keywords were revised.

Point 7:     Please check the formatting and editing according to the journal's requirements. This also includes the reference and bibliography.

Response 7: checked and edited according to the journal's requirements.

Point 8:     Please cite more relevant literature in the manuscript from recent years (2022-2023).

Response 8: Recent literature published in the last two years were added.

Point 9:     Please mention the reason to select Seven accessions for drought stress and twelve accessions for high-temperature stress.

Response 9: As described in the manuscript, the tested accessions were selected from our previous published studies in which a smaller number of accessions were tolerant to drought compared to heat stress.

Point 10:     Please include pictures of plants during stress treatment or controlled conditions.

Response 10: Unfortunately, the pictures we took were of poor quality. For that reason, we did not include them in the manuscript.

Point 11:     Please include genetic references for the role of biochemical and enzymatic features in stress tolerance in lentils or other plants.

Response 11: Genetic references were added.

Point 12:     Please mention the identities of better-performing accessions in the abstract.

Response 12: the top performing accessions were mentioned in the abstract.

Point 13:     Line 47: Define the “range of abiotic stresses” to enhance clarity.

Response 13: the range of abiotic stresses was defined.

Point 14:     Line 49: “damage to agricultural crop growth” can be rewritten for better understanding.

Response 14: the sentence was improved.

Point 15:     Line 52-53: Rephrase the line to enhance clarity.

Response 15: the sentence was improved.

Point 16:     Line 59: Capitalize the first letter of the words at the beginning of a sentence.

Response 16: Done

Point 17:     Line 59: Replace "holds the significant position of being" with "holds a significant position as the".

Response 17: Done

Point 18:     Line 62: Rewrite the sentence to provide a better understanding of the concept to the reader.

Response 18: the phrase was improved.

Point 19:     Line 66: Change "diminishing its production and productivity" to "diminishing its yield and productivity".

Response 19: Done

Point 20:     Line 68: Correct "receiving increased attention due to their substantial threat" to "receiving increased attention due to the substantial threat".

Response 20: corrected.

Point 21:     Line 94: What do you mean by “high temperature”? can you mention the numbers?

Response 21: the number was mentioned.

Point 22:     Line 101: high temperatures stress (32/20 °C)

Response 22: corrected.

Point 23:     Line 394: You mentioned the “anthesis phase” Kindly define it when using it for the first time. 

Response 23: Information was provided.

Reviewer 2 Report

I carefully read the submission titled “Enzymatic and Biochemical Reactions of Lentil (Lens culinaris 2 Medik.) to High Temperatures and Drought Stresses in the Re-3 productive Stage”. My first impression that the paper contains new information and title of the manuscript cover its content. The summary is appropriate and the aim of the work clearly established. The methods are used are adequate and used sophisticated techniques and equipment's. I found the results very reliable. However, introduction, results and conclusions are not well documented and scientifically coherent.

Therefore, I have some corrections and additions on it before acceptance.

My suggestions and corrections are shown on attached manuscript file.

Author Response

To: Ms. Marijana Apic

Journal Name: Plants (ISSN 2223-7747)

Special Issue: Abiotic Stress-Induced Secondary Metabolites Regulating Plant Metabolism

Section: Plant Physiology and Metabolism

Date: August 17, 2023

Revision for the Manuscript ID: plants-2557895

Dear Editor,

We appreciate you and the reviewers for your precious time in reviewing our paper and providing valuable comments. We have carefully considered the comments and tried our best to address every one of them. We hope the manuscript after careful revisions meet your high standards. The authors welcome further constructive comments if any.

Below we provide the point-by-point responses to the comments of the reviewer 2 alongside this letter. Track changes was used to highlight the changes in the manuscript, these changes have been approved by all co-authors.

Sincerely,

Dr. EL HADDAD Noureddine

Point 1:     Some important numerical findings should be added to the abstract. The summary section is written like the conclusion section. Therefore, the summary part is weak and should be rewritten.

Response 1: The abstract was improved, and more information about our key findings was added.

Point 2:     Information about phenolic compounds should be given. Literature on the relationship between temperature stress and phenolic compounds should be added.

Response 2:  This information was added in the manuscript with references to recent literature.

Point 3:     There are deficiencies in the interpretation of the findings obtained in the Results section. Findings should be given by interpreting rather than giving them directly. This will help to inform the readers more about the results of the research.

Response 3: This section was revised and improved.
